# A snapshot of biodiversity protection in Antarctica

Hannah S. Wauchope[1], Justine D. Shaw[2] & Aleks Terauds[3]

Threats to Antarctic biodiversity are escalating, despite its remoteness and protection under the Antarctic Treaty. Increasing human activity, pollution, biological invasions and the omnipresent impacts of climate change all contribute, and often combine, to exert pressure on Antarctic ecosystems and environments. Here we present a continent-wide assessment of terrestrial biodiversity protection in Antarctica. Despite Antarctic Specially Protected Areas covering less than 2% of Antarctica, 44% of species (including seabirds, plants, lichens and invertebrates) are found in one or more protected areas. However, protection is regionally uneven and biased towards easily detectable and charismatic species like seabirds. Systematic processes to prioritize area protection using the best available data will maximize the likelihood of ensuring long-term protection and conservation of Antarctic biodiversity.

[1] Department of Zoology, University of Cambridge, The David Attenborough Building, Pembroke Street, Cambridge CB2 3QZ, UK. [2] School of Biological Sciences, The University of Queensland, St Lucia, QLD 4067, Australia. [3] Antarctic Conservation and Management Program, Australian Antarctic Division, Kingston, TAS 7050, Australia. Correspondence and requests for materials should be addressed to A.T. (email: aleks.terauds@gmail.com)

Antarctica is often viewed as a wilderness untouched by the threats facing the rest of the world, but recent work has shown that conservation trajectories in Antarctica are similar to those globally[1]. Threats to Antarctic ecosystems and biodiversity are increasing—primarily from climate change, biological invasions, pollution and the increasing footprint of human activity[2–6]. Antarctica has had a history of unique continent-wide protection, first through the Agreed Measures for Conservation of Antarctic Flora and Fauna (1964)[7] and more recently under the Protocol on Environmental Protection to the Antarctic Treaty (hereafter the Protocol)[8], which came into force in 1998 and designates Antarctica as a 'natural reserve, devoted to peace and science'. This Protocol provides a high level of environmental protection, including a ban on mining and mineral exploration, prohibition of the intentional introduction of non-native species, strict regulations on disturbance to native species, waste management and environmental impact assessment requirements. However, even under this unique level of protection, dramatic impacts on Antarctic ecosystems have been documented, particularly near high concentrations of human activity[4,9]. In recognition that additional protection is sometimes necessary, Annex V to the Protocol (which came into force in 2002)[8], also provides for Antarctic Specially Protected Areas (ASPAs), which can be designated to protect outstanding environmental, scientific, historic, aesthetic or wilderness values and/or scientific research, or a combination of these values (Supplementary Table 1).

ASPAs are fundamental to reducing human impacts on Antarctic ecosystems and biodiversity. They require a management plan that details the permitted activities and provides visitation guidelines and in all cases entry into ASPAs is only allowed with a permit from a designated national authority. Importantly, once an ASPA is adopted by consensus by the Antarctic Treaty Consultative Parties, it is the joint responsibility of all signatories to the Protocol to ensure that the values within the area continue to be protected[10]. ASPAs mitigate threats, largely through a significant reduction in human activities (both science and tourism) and a concomitant decrease in associated impacts from biological invasions and environmental degradation. While climate change impacts may not be directly mitigated by increased protection of this nature, the alleviation of other major threats will reduce synergistic impacts (e.g. see discussion in ref. [11]). Protected areas have been shown globally to maintain species population levels better than other management approaches and reduce rates of habitat loss[12], and several recent studies have also highlighted that this is also likely to be the case in Antarctica[13–17].

There is growing discussion in both global policy forums (e.g. in the Antarctic Treaty system's Committee for Environmental Protection[18]) and scientific literature[14,16,17] on the need for ASPAs to address future threats. There is also increasing scrutiny of the current suite of ASPAs regarding their adequacy, representativeness and proximity to human activities[10,14–17]. However, the extent to which Antarctic species are protected by ASPAs is not known. Clearly, defining this is critical. Annex V of the Protocol calls for ASPAs to be identified "within a systematic environmental-geographic framework"; quantifying the representation of biodiversity within existing protected areas is not only consistent with this call, but crucial to supporting further discussions and improving area protection in the face of increasing human activities in Antarctica[14,16,17].

Here, we present an overview of biodiversity protection in terrestrial Antarctica, at both a continental and regional scale, utilizing the most comprehensive Antarctic biodiversity dataset yet compiled—containing over 48,000 species records. We underpinned these analyses with an updated spatial layer that accurately defines the ASPA locations and associated boundaries, and spatially explicit species occurrence records extracted from each ASPA Management Plan[19] (updated ASPA layer and ASPA biodiversity records available from data.aad.gov.au)[20,21]. We quantify the species found within ASPAs, identify taxonomic biases in coverage, and assess regional species protection. Despite deficiencies in the current coverage of biodiversity by ASPAs, the Antarctic Treaty system provides a solid foundation to improve biodiversity protection across the continent.

## Results

**The data.** A comprehensive compilation of Antarctic biodiversity data is problematic due to the continent's remoteness and the inaccessibility of many locations (see also related discussion in ref. [22]). Furthermore, the level of biodiversity knowledge within ASPAs also varies, with some ASPAs being more comprehensively surveyed that others. Some of these issues could be overcome using species distribution models (e.g., see refs. [23,24]) but given the complexities in these types of analyses (e.g., ref. [25]), including modelling large number of separate taxa (over 2000 species in the database), overcoming spatial autocorrelation effects due to the fragmented nature of the ice-free landscape, and combining the distributions to form a meaningful output (e.g., see ref. [26]), we have limited the scope of our analyses to occurrence data. In our view, such data are more appropriate to use as a first step in providing a spatially explicit and species-specific overview of biodiversity protection, which can then be built and/or expanded on through the use of modelled or proxy data. Our database is more comprehensive than anything yet compiled for terrestrial Antarctica, and includes records from major Antarctic herbaria, hundreds of scientific publications and all ASPA management plans. In this context it provides more than sufficient information to achieve our primary objective—to inform and progress continent-wide, systematic area protection in terrestrial Antarctica.

**Biodiversity coverage.** Terrestrial biodiversity in the Antarctic is predominantly restricted to areas that are permanently ice-free—currently estimated at somewhere between 0.2 and 0.5% of the Antarctic continent (21,745 km[27] to 45,886 km[28]). We found that 44% of species (including birds, plants, lichens and invertebrates, but not microbes or marine species), occur within ASPAs. This is surprisingly high given only 1.5% of ice-free area is protected[16,28]; however, we find that for species occurring in ASPAs, 52% only occur in one ASPA. All vertebrate species considered here (all 21 of which are birds—see Methods) were found in at least one ASPA, which is consistent with other large scale analyses that show vertebrates, and particularly birds, typically have higher levels of protection (e.g., see refs. [29,30]). By contrast, plants and lichens, a group predominantly made up of algae, mosses, liverworts and lichens, only have approximately one-third of species (500 of 1216 species) in ASPAs while invertebrate species fare only slightly better, with just over half of the species (154 of 302 species) found in at least one ASPA.

**Regional protection.** Antarctica has 16 biologically distinct ice-free regions (Antarctic Conservation Biogeographic Regions—ACBRs or ecoregions hereafter), defined by their biological communities and underlying physical, climatic and environmental characteristics (Fig. 1a, Supplementary Figure 1, Supplementary Datasets 1,2,3, Supplementary Software 1)[28,31]. These ecoregions provide a useful spatial framework for representing the diversity of species and communities across Antarctica, particularly where finer resolution sampling of biota may be lacking. The protection of biodiversity within ASPAs is uneven across the 16 ecoregions (Fig. 1a, Supplementary Figure 1). Four do not contain any ASPAs[16,28] and in those where ASPAs are present,

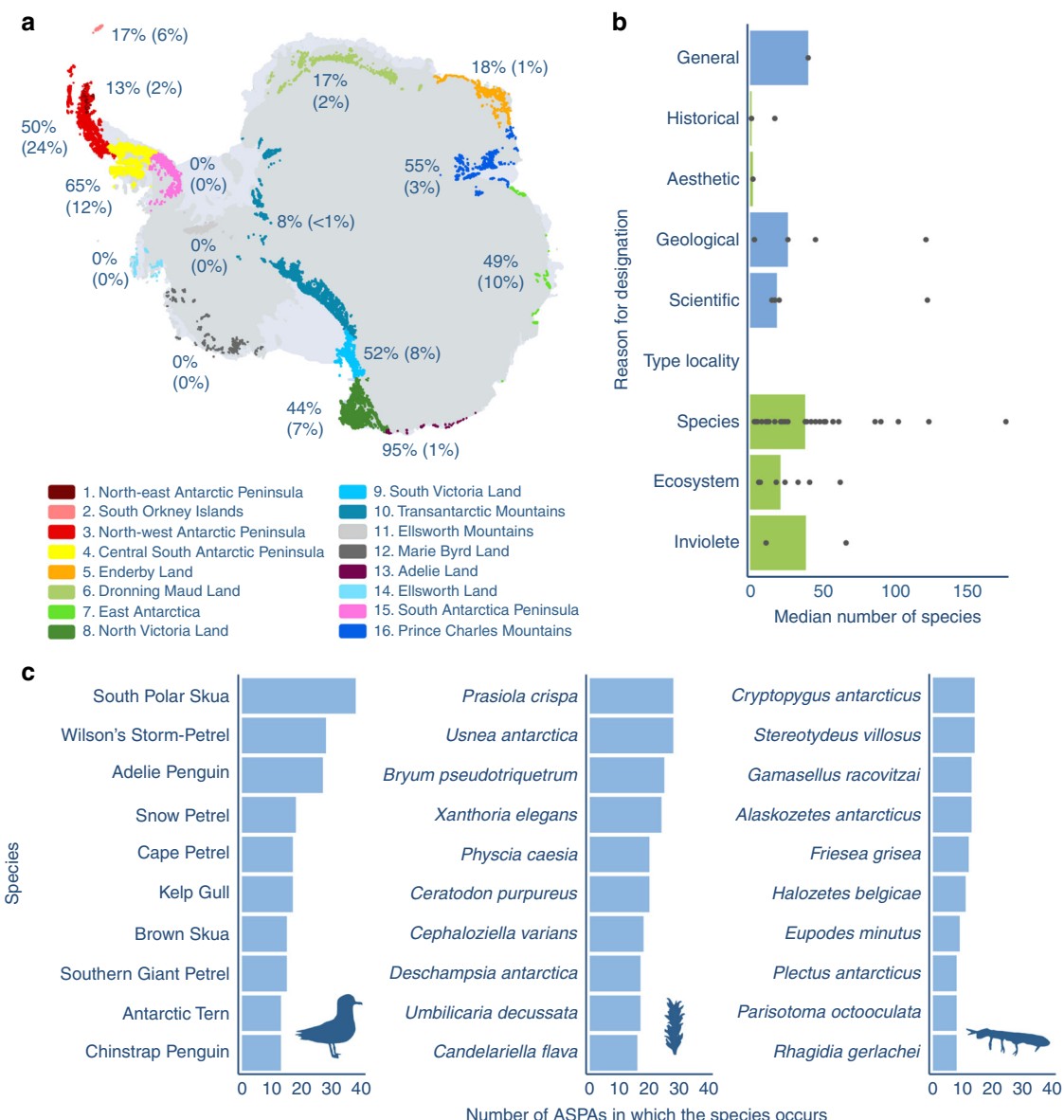

**Fig. 1** A snapshot of Antarctic biodiversity protection. **a** Antarctic ice-free areas are split into 16 Antarctic Conservation Biogeographic Regions[28,31] (ACBRs—denoted by the different colours). The % of species in each ACBR that occur in Antarctic Specially Protected Areas (ASPAs) is shown in bold. The % of all Antarctic species that occur in ASPAs within each ACBR is shown in brackets. **b** ASPAs are designated for a range of reasons (y-axis). Bars show median number of species in ASPAs that have been primarily designated to protect biodiversity related values (green) or non-biodiversity values (blue), with points for number of species in ASPAs of each designation overlaid. **c** The ten vertebrates (left) plants and lichens (middle) and invertebrates (right) that are found in the greatest number of ASPAs. The Antarctic map (1a) uses the coastline layer from the SCAR Antarctic Digital Database (Version 7 www.add.scar.org) and the ACBR layer from the Australian Antarctic Data Centre[38]. Data and code used to create these figures are provided as Supplementary Datasets 1,2,3,5 and Supplementary Software 1

the proportion of species occurring in at least one ASPA ranges from 8-95% (Fig. 1a). However, the uneven distribution of taxa across the continent (Supplementary Figure 2, Supplementary Datasets 1,2,4 and Supplementary Software 1) means that an ecoregion with a high proportion of species in ASPAs may not necessarily protect a high proportion of all Antarctic biodiversity. These analyses are also influenced by the different areas of ASPAs across ecoregions, as some of the larger ASPAs in smaller ecoregions (e.g. ASPAs 147 and 170 in ACBR 4) contain a relatively large number of species (see also Supplementary Figures 2 and 3, Supplementary Datasets 1,2,3,4,5, Supplementary Software 1).

Using satellite data, Hughes et al.[14] found that protection of vegetation within ASPAs was greatest in the South Orkney

Islands and the North Antarctic Peninsula ecoregions. Similarly, we found ASPAs in the North-West Antarctic Peninsula ecoregion protect the second largest proportion of species (50, and 24% of all Antarctic species). However, in contrast, we found that ASPAs in the species rich South Orkney Islands included proportionally fewer species than the average ecoregion (17 c.f. 30%). The highest level of representation of species in ASPAs was found in the Adélie Land ecoregion, where 95% of species were found in at least one ASPA; however this represents only 1% of all Antarctic biodiversity, due to the small number of taxa known to be present in this ecoregion (Supplementary Figures 1 and 2).

**Biodiversity vs non-biodiversity protected areas.** Assessing the actual level of protection afforded to species through their

occurrence within ASPAs is complicated by the reason for which each ASPA was designated[32] (see Supplementary Tables 1,2 and Supplementary Dataset 6 for designation details). In some cases, biota found within non-biodiversity designated ASPAs (e.g. one that has been designated to protect historical infrastructure or scientific values) may be afforded little to no more protection than those outside of the ASPA, as the ASPA's Management Plan is not designed to mitigate potential impacts on these species. For example, scientists undertaking geological research have inadvertently trampled rare plant communities in ASPA 140, which was designated to protect scientific values[32]. In other cases, ASPAs are designated to protect specific species and while the Management Plans may specify provisions to protect those species, they may offer little or no protection to other species found within the area[32]. While there is provision for the management of multiple values under the Protocol, there are only a few examples within the current suite of ASPAs where this has been explicitly attempted (e.g. ASPA 123 was designated for its 'outstanding aesthetic and wilderness values' but is also managed to protect biodiversity values)[19].

While most species found in ASPAs are located within ASPAs designated for biodiversity, the distribution of protection across designations is uneven, and 9% of species are found only in non-biodiversity designated ASPAs (Fig. 1b, Supplementary Figure 4, Supplementary Datasets 1,2,3,4 and Supplementary Software 1). Furthermore, five ecoregions have no ASPAs that were designated for biodiversity protection and if only biodiversity ASPAs are considered, the mean number of species that occur in at least one protected area drops to 40% (c.f. 44% for all ASPAs regardless of designation). These findings highlight the opportunity for proponents of ASPAs to consider managing for multiple objectives, ensuring appropriate management for the original designation, the associated biodiversity, and any other values that are present (see also discussion in ref. [32]). Such an approach is also consistent with Annex V of the Protocol, which explicitly states that ASPAs can be designated to protect a combination of values[8].

**Species specific occurrence in protected areas.** Overall, the species found in the greatest number of ASPAs across the taxonomic groups were south polar skuas *Catharacta maccormicki*, the algae *Prasiola crispa* and lichen *Usnea antarctica*, and the springtail *Cryptopygus antarcticus* and mite *Stereotydeus villosus*. Two of the four Antarctic penguins (Adélie penguins - *Pygoscelis adeliae* and chinstrap penguins - *P. antarcticus*) were in the top 10 vertebrates, and most of the top 10 plants and lichens were lichens (6) or mosses (2) (Fig. 1c, Supplementary Datasets 1,2,3,5, Supplementary Software 1). Of the two native Antarctic vascular plants, only the grass *Deschampsia antarctica* was in the top 10 plant and lichen species. The top 10 invertebrates were dominated by mites (6) and springtails (3), with one nematode species. None of the top 10 invertebrate species occur in more ASPAs than any of the top ten vertebrates (Fig. 1c), highlighting the disparity between these two groups. There is considerable variability in the spatial extent of taxa across the continent[31], which is likely to explain some of these differences both within and between broad taxonomic groups (see also Supplementary Figures 1 and 2). We also assessed the top 10 species based on the areas of ASPAs in which they are found—with broadly similar results for vertebrate and plants and lichens, and five new species, including a higher predominance of nematodes, in the invertebrate group (Supplementary Figure 3).

## Discussion
While the database underpinning these analyses is the most comprehensive yet compiled for the terrestrial Antarctic region, it does have limitations. We do not include an assessment of ASPAs that are predominantly marine, largely due to a lack of available data in nearshore ecosystems, both at a continental scale and within ASPAs. Better understanding the distribution of nearshore biodiversity, and the protection it is afforded, warrants more attention, particularly given the burgeoning interest by some Antarctic Treaty Parties in harmonizing area protection across nearshore and pelagic realms[33]. We also did not include microbial life in our assessment, again due to a lack of taxonomically robust and comprehensive data. This is an area that would also benefit from further research, as Antarctic microbial communities are increasingly being recognized as vulnerable to human impact[34,35].

Another limitation of the underlying data is uneven survey effort across ecoregions (Supplementary Figure 1) and ASPAs, and there are also biases in survey efforts across taxa[22]. In particular, some ASPAs may be better surveyed than other, non-protected areas, potentially biasing the representation of some species. Some of these biases could potentially be overcome using Species Distribution Models, and this is likely to represent a productive area of future research in the assessment of existing protection, and the identification of important new areas. Nevertheless, and despite these limitations, the snapshot of biodiversity protection presented here is an important advance in understanding the overall coverage of species by formal area protection in Antarctica, and highlights a bias in current ASPAs towards protecting easily detectable and charismatic species over less visible species, many of which are the predominant terrestrial biodiversity[22]. Furthermore, this snapshot provides a foundation for the systematic development of area protection via the mechanisms provided for under the Antarctic Treaty system, particularly Annex V to the Protocol, including the protection of multiple values by individual ASPAs.

Globally, assessing the efficacy of area protection is progressing rapidly. Recent studies have shown that care is needed when developing protected area systems, as too much of a focus on protecting a certain amount of area can lead to perverse outcomes[36]. Similarly, situating protected areas where they will not impact competing activities can also lead to ineffectual protection[37]. Here we provide an important spatial-specific and species-specific assessment to inform and assist targeted continent wide systematic area protection for terrestrial Antarctica. Our assessment clearly identifies the lack of representativeness of protected areas both at a regional and species level. This is somewhat unsurprising, given that a clear understanding of the spatial distribution of Antarctic terrestrial biodiversity has only recently been available. This situation can be readily addressed given the apparent international will, through the mechanisms available through the Protocol, and informed by systematic conservation planning tools (e.g., see ref. [17]). Systematic processes to prioritize area protection, based on the best available data, will maximize the likelihood of meeting the objectives of the Protocol on Environmental Protection to the Antarctic Treaty and ensuring long-term protection and conservation of Antarctic biodiversity.

## Methods
**Overall approach.** We proceeded in three clearly defined stages: (1) Digitize and consolidate spatially explicit data on Antarctic biological life. (2) Assess and georectify the polygons representing the ASPAs (building on the work of Terauds and Lee[28,38]). (3) Summarize the protection of Antarctic biodiversity on a continental and regional scale.

**The SCAR Antarctic Biodiversity Database.** We built upon the SCAR Antarctic Biodiversity Database compiled and used by Terauds et al.[31], by searching for and

adding spatially explicit records from the published literature. We found data in 178 sources, adding 7364 records comprising 440 species to the original database used by Terauds et al.[31] We then extracted all spatially explicit biodiversity data reported in the 72 existing ASPA management plans[19] adding a further 2438 records to the database (these records are available at data.aad.gov.au)[21]. Where possible we quantified the uncertainty around these spatial locations, and documented the original source of the record[21]. The final biodiversity database used here holds 48,602 records, 2292 unique species and represents the most comprehensive consolidation of Antarctic terrestrial biodiversity information yet compiled.

**Antarctic specially protected area spatial layers.** Starting with the most recently updated polygon shapefile of ASPAs[38], which contained some minor improvement on the original ASPA spatial layer first made publicly available in 2011[39], we first cross-checked the location of ASPA polygons with the spatially explicit locations provided in the ASPAs Management Plans[19]. Boundary points in the ASPA polygons were moved to align with the coordinates in the Management Plans, unless a clear error was identified. This initial cross checking highlighted several issues with some polygons, and approximately 1/3 of ASPA polygons with boundary coordinates provided in the Management Plans had at least some errors, equating to almost 10% of all boundary points being identified as incorrect.

Once polygons were aligned with the Management Plans, we then georeferenced the maps provided in the management plans to check the ASPA boundaries in relation to known landscape features. We used the most recent update of the permanently exposed rock layer[27] to locate these features as it is currently considered the most accurate (and conservative) representation of these permanently exposed ice-free areas[27]. If Management Plan boundary coordinates did not match with the rock and coastline features (coastline from Antarctic Digital Database version 7 www.add.scar.org), they were checked in Google Earth to assess whether the coordinate or the feature layer was incorrect. Most ASPA polygons needed some adjustment to align with the rock and coastline, with distances moved ranging from 1 m to ~10 km (~800 m on average). Ten of these ASPAs had only very minor adjustments (<20 m). In two cases the coordinate did not match rock or coastline layers but did fall on rock according to Google Earth; in these cases, the coordinates were not changed. Central points (which are provide as a separate shapefile) were updated to ensure they always fell within the ASPA, this required shifting away from the true centre for irregularly shaped ASPAs. Snaps to coastline were only conducted when this increased the resolution of the ASPA rather than decreasing it. In some cases, there was a lack of concurrence between co-ordinates, PDF map, coastline, rock layer or Google Earth. In these cases the following protocol was followed: snap to coordinates (unless clearly wrong), otherwise align to rock outcrop layer based on the PDF map, otherwise align to coastline. Full details of the updates made to each ASPA can be found in the README file accompanying the updated layer. The updated version is publicly available through the Australian Antarctic Data Centre (data.aad.gov.au)[20].

**Data preparation.** First, any species records with a spatial uncertainty of more than 10 km were removed, as were points that were shifted more than 10 km to snap to the rock outcrop shapefile. All marine and ice breeding species were removed, with the exception of emperor penguins (*Aptenodytes forsteri*), which were included on the basis of the comprehensive information on their breeding locations (e.g. see http://www.penguinmap.com/Species/) and the fact that several ASPAs were designated specifically to protect this species. Antarctic fur seals (*Arctocephalus gazella*), southern elephant seals (*Mirounga leonina*) were also excluded as data for these species were not compiled as part of the SCAR Antarctic Biodiversity Database. Microbial taxa (primary from the Kingdoms Bacteria, Chromista and Protozoa) were also removed due to lack of taxonomically robust and comprehensive data.

**Protection by taxa/species.** We categorized the biota into three broad taxonomic groups:

(i) Vertebrates—comprised of penguins (four species), petrels (six species) and other bird species.
(ii) Plants and lichens—defined by all taxa of Kingdom Plantae and Division Ascomycota within Kingdom Fungi; largely algae, mosses, liverworts and lichens.
(iii) Invertebrates—defined by all animals that are not chordates; including tardigrades, mites, springtails and nematodes.

For Fig. 1 we calculated the number of protected areas each species occurred in. For Supplementary Figure 3, we calculated the total area of ASPAs each species occurred in.

**Regional protection.** For regional protection we first quantified the total number of species occurring within each ACBR by overlaying species records with the ACBR layer. We then quantified the percentage of these occurring in ASPAs by overlaying records with the ASPA layer; providing the percentage of species in each ACBR that fell in at least one ASPA. For Supplementary Figure 1, we considered how many of the protected species only occurred in non-biodiversity designated

ASPAs, shown in purple. To check for the influence of sampling bias, we considered how much of each ACBR had been sampled, by converting each ACBR polygon into $1 \times 1$ km grid cells, and calculating the percentage that contained at least one biodiversity record. Finally, we assessed the area of each ACBR covered by ASPAs, by intersecting each ACBR with any ASPAs falling within it, and showing the area covered by ASPAs designated for biodiversity (green) and other reasons (purple) (Supplementary Figure 1d).

**Protection by designation.** Each ASPA has a specific reason for its designation (Supplementary Table 1) and we assigned each of these to "biodiversity" or "non-biodiversity" categories, based on the details of the criterion assigned to the ASPA (Supplementary Table 2). All ASPAs of criterion B and C were assigned as biodiversity ASPAs. Those of A, E, G and I were assessed on a case by case basis, as those criteria may or may not include biodiversity protection. Criterions F and H were assigned non-biodiversity status.

We then quantified the number of species that were protected by ASPAs, the proportion of these that only occur in non-biodiversity ASPAs and the median number of species protected by ASPAs of each designation. Plots of the summed number of species protected by ASPAs of each designation, as well as plots showing the number of species in each ASPA, and finally a plot of the total number of ASPAs of each designation are provided in Supplementary Figure 4.

**Reporting summary.** Further information on experimental design is available in the Nature Research Reporting Summary linked to this article.

## Data availability

A reporting summary for this Article is available as a Supplementary Information file. The spatially explicit biodiversity data from the Antarctic Specially Protected Area Management Plans are publicly available from the Australian Antarctic Data Centre (https://doi.org/10.26179/5c1b10c534c19). The revised and updated Antarctic Specially Protected Area spatial layers (points and polygons) are also available from the Australian Antarctic Data Centre (https://doi.org/10.26179/5c1b15eedaeb6). The SCAR Antarctic Biodiversity Database, which underpinned the comparative analysis, is currently being prepared for submission as a data paper, and as such is not publicly available at this time. Once published these data will also be publicly available from the Australian Antarctic Data Centre. In the interim please contact the corresponding author for access to this dataset. Source data underlying Fig. 1a–c, and Supplementary Figures 1–4 are provided as Supplementary Datasets and referenced in each Figure legend. A detailed description of these datasets is provided in Supplementary Dataset 1.

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

## Acknowledgements

This research was supported by the Scientific Committee on Antarctic Research (SCAR) Standing Committee on the Antarctic Treaty System and contributes to the SCAR Scientific Research Program – State of the Antarctic Ecosystem. A.T. is supported by the Australian Antarctic Program – Project 4296. We thank E. McIvor and P. Tracey for helpful suggestions on earlier versions of this manuscript, and M. Wauchope and B.I. Simmons for helpful comments on figures.

## Author contributions

A.T. initiated the study and is currently the primary custodian of the SCAR Antarctic Biodiversity Database. H.W. digitized biodiversity records from ASPAs, added extra records to the database from published sources, updated the ASPA layer, and led the analyses with contributions from A.T. and J.D.S. H.W. conceived and created the figures with input from A.T. and J.D.S. All authors co-wrote the paper.

## Additional information

**Competing interests:** The authors declare no competing interests.

