## [Peer Review File · Nature Communications]

Reviewers' comments:

Reviewer #1 (Remarks to the Author):

A snapshot of biodiversity protection in Antarctica
Wauchope et al.

Reviewed by Kevin A. Hughes, British Antarctic Survey, Cambridge, UK.

General comments

The authors have tackled a question that is one of the major factors holding back progress with the development of the Antarctic Protected Area System through the Antarctic Treaty System's Committee for Environmental Protection. The conclusions represent a significant and major advance in the field that will be of great benefit to scientist and policy makers within the 40 nations that have signed the Protocol on Environmental Protection to the Antarctic Treaty. Annex V to the Protocol, which describes the mechanism and criteria for area protection, is rather vague and which has led to inconsistencies in the development of the ASPA network. Added to this, the majority of ASPAs were designated before Annex V entered into force, previously being SSSIs or SPAs. This has led to a very confused system which makes analysis difficult, and inevitably the authors have had to make some judgement calls during this research which may be open for debate. Nevertheless, Wauchope et al. have used a sensible and carefully considered methodology that has moved the whole areas forward very significantly. I have made a few comments below, suggesting areas where their analysis might be re-examined, but overall I see this work as substantially progressing the field and I look forward to seeing it published.

Specific comments

Page 1 Line 18: It might be helpful to briefly mention which threats to Antarctic biodiversity are escalating.

Page 1. Line 30-31. Antarctic had protection before 1998 through the Agreed Measures for Conservation of Antarctic Flora and Fauna (1964). Much of the Protocol built on the Agreed Measures. Also, Annex V did not enter into force until 2002 (not 1998).

Page 2 Line 65. Semi-colon before 'however'

Page 3 Line 67 and Methods Page 7 Line 207-211. I wonder if the authors can explain in more detail their rationale for excluding seals and emperor penguins from the study, particularly, for example, as ASPA 137 Northwest White Island was designated with the primary object of protecting the Weddell Seal colony located there and several ASPA (e.g. APSAs 101, 107, 127, etc.) have been designated primarily to protect emperor penguins. The well-documented breeding locations of these species on ice, might mean that their inclusion is possible.

Page 3 Line 69-72. I found it interesting that fungi were included in the analysis. I am not sure that the data from more recent molecular biodiversity work is routinely incorporated into the Antarctic databases used for this analysis. Vascular plants, bryophytes and lichens have probably been characterised to the greatest extent. However, existing fungal diversity records often rely upon earlier culture work and may not take into account the much wider diversity detected by, for example, environmental DNA methodologies. I wonder how confident the authors are that this analysis of fungi is fully representative of the available data.

Page 3 Line 86-88. The size, rather than the number of ASPAs in a particular ACBR may put a different perspective on the results shown (i.e. see Figure 1d and Supplementary Figure 1a and c). For example, the limited number of ASPAs in ACBR 4 are large in size compared to most ASPAs,

and are also the location where much of the biodiversity research has been done. This may explain the high degree of species known from the ACBR that are found within ASPAs. However, other areas within this ACBR have been little explored and the overall biodiversity of the ACBRs is likely be greater. The authors have described some of these issues on page 4 lines 121 onward, but perhaps some of the complexities could be at least mentioned earlier in the manuscript, to give context to the examples provided in this example.

Page 3 Line 99 to Page 4 Line 107 and Supplementary Table 2. Looking at the list in Supplementary Table 2, I think the author may have been a little simplistic with their allocation of 'Protocol designation', and not perhaps adequately recognised that multiple values can be protected within an ASPA. The authors could consider re-labelling this column 'Primary Protocol Designation'. This also relates to how the information is shown in Figure 1a. I'm not sure this does justice to ASPAs protecting a combination of values.

The authors could also add a column detailing in which ACBR the ASPA is found.

I note that the authors make little mention of the ASPAs that protect predominantly marine biodiversity. It might be useful to explain a little more how they fit (or not) into this study. Is this an opportunity for further work?

I would also suggest that the authors check a few of the ASPAs that have been described as not being designated for protection of biodiversity (with '0' in column 4). I think that ecological values have been included in some of these ASPA. I have provided some text from the relevant ASPA Management Plans for some examples, which show that biological values are part of the reason for designation:

- ASPA 123 Barwick and Balham Valleys ' Some of the best examples of the physical surface features associated with this unique and extreme environment are found on the valley floors, where there are also fine examples of microbial life, lichens, as well as soil and lake microflora.'
- ASPA 172. Lower Taylor Glacier and Blood Falls. The primary reasons for designation of the Area are its unique physical properties, and the unusual microbial ecology and geochemistry.
- ASPA 147. Ablation Valley and Ganymede Heights. The primary reason for the designation of Ablation Valley and Ganymede Heights, Alexander Island (70°48'S, 68°30'W, approximately 180 km²) as an Antarctic Specially Protected Area (ASPAs) is to protect scientific values, relating particularly to the geology, geomorphology, glaciology, limnology and ecology of this extensive ablation area.

Page 4 Line 109-111. I note the emphasis has been place on the presence of species in the greatest 'number' of ASPAs (see earlier comment). Rather I wonder if it is possible to take ASPA areas into consideration. For example, ASPA 126 Byers Peninsula has an area of 90 km² and ASPA 147 Ablation Valley has an area of 110 km². These values contrast greatly with the much smaller areas of many other ASPAs, such as ASPA 110 Lynch Island with an area of only 0.15 km². Hence some mention of ASPA area would be important and give more accurate representation of the relative level of protection of some species.

Page 4 Line 130. Change to 'Annex V to the Protocol'

Page 5 Line 136-137. Here the authors write 'Here we provide an important spatial- and species specific assessment to inform and assist targeted continent-wide systematic area protection for terrestrial Antarctica.' I agree fully with this assessment and would suggest that this stated earlier in the manuscript.

Page 5 Line 141-146. Not necessarily for inclusion here, but the authors may like to consider the work of Hughes and Grant which is currently in press in Polar Research entitled: 'Current logistical capacity is sufficient to deliver the implementation and management of a representative Antarctic protected area system'

Page 5 Line 162. I think there are 72 ASPAs, not 73, i.e. number 101 to 175 (75), minus ASPAs 114, 118 and 130 (3) gives 72 in total.

Page 6 Lines 180-201. The work to resolve the boundary issues is of great value as this has been an issue that had caused difficulties for scientists and policy makers in the past. The approach taken seems reasonable, given the complexities.

Page 7. Lines 226-228. The authors should recheck some of the ASPAs that have been described as 'non-biodiversity designated ASPAs', e.g. ASPAs 147, 123, and possibly amend the size of the 'purple' bars as appropriate in Supplementary Figures 1b and 1d. This may also lead to tweaks of Supplementary Figure 2 and part of Figure 1.

References.

The references listed seem reasonable and appropriate for this research area.

Supplementary Table 1. The Environmental Protocol in Annex V states: 'Any area, including any marine area, may be designated as an Antarctic Specially Protected Area to protect outstanding environmental, scientific, historic, aesthetic or wilderness values, any combination of those values, or ongoing or planned scientific research.' I think it is important that this broader concept of multiple values being protected in a single ASPA is highlighted, particularly in light of the statements made on Page 4 Line 104-107, i.e. 'These findings highlight the opportunity for proponents of ASPAs to consider managing for multiple objectives, ensuring appropriate management for the original designation, the associated biodiversity, and any other values that are present.'

I note that this work is supported by the SCAR Standing Committee on the Antarctic Treaty System, which is the conduit for communication between SCAR and the Antarctic Treaty System, including the Committee for Environmental Protection, which oversees ASPA management plan preparation. Based on the comments here, I wonder if SCAR would be willing to provide biodiversity data, revealed in the research, for new ASPAs, or during the revision of existing ASPA management plans.

Reviewer #2 (Remarks to the Author):

Overall

I find this manuscript to be well written and the analysis methods simple and easy to understand. I was surprised that the authors don't take the opportunity to provide specific advice about where the next best places might be to place protected areas. I was also surprised to see the authors use only point records of biodiversity as a means of assessing the ASPAs given that this method of evaluating and designing conservation reserves appears to have some significant limitations compared with using modelled distributions (Wilson et al. 2004; Rondinini et al. 2006). This strategy needs to be clearly justified against alternatives, including utilising modelled species distributions. Clearly, there could be some species present in ASPAs that have not been observed there yet, and there could be some species present in eco-regions that have not been recorded in those regions yet. This needs to be at least acknowledged, if not explicitly dealt with in analysis.

Generally, the point/value of ASPAs was not particularly well introduced. Given that Antarctica has "had unique continent-wide protection since 1998, under the Protocol on Environmental Protection to the Antarctic Treaty", it is hard to feel strongly motivated by the representativeness or otherwise of the ASPAs. Especially when it appears that a bunch of them are not actually aimed at protecting biodiversity and even when they are, it is not clear what actually happens on them that

doesn't happen elsewhere (or vice-versa). How many of the species in your analysis are faced with a higher risk of extinction because they're not in ASPAs?

To that end, I was left feeling unclear about whether the representativeness is really a very good way to assess conservation effort/outcome on Antarctica. Some assessment of adequacy, referred to early in the manuscript would seem very appropriate, but that is not currently provided. If climate change is the top driver of future biodiversity loss in Antarctica, then how will ASPAs help?

Specific comments

Line 18 – is it all of biodiversity (including insects and bryophytes?) or just vertebrates – mention here so readers can get an idea before diving into the paper.

Lines 36-40 – Given the apparently strong protection of biodiversity on Antarctica, it is unclear what additional protection ASPAs provide. Therefore it is hard to feel strongly motivated by the need to see how representative these ASPAs are in terms of biodiversity. Can you motivate this story a bit more with something about the biodiversity impacts that are mitigated/avoided on ASPAs? (and how?)

Line 49. Vague. What discussions? Any in particular? What do you mean by area protection? Could you refer to any international agreements here?

Line 55 – what is an ASPA spatial planning layer? What does it contain? Why would that be interesting?

Line 62 – This statement doesn't seem to fit with the rest of the paragraph. Why would the perceived 'simplicity' of Antarctic ecosystems be relevant to the rest of this paragraph which seems to be about where the biodiversity is and how well protected it is. "While Antarctic terrestrial ecosystems have often been considered relatively simple, their structure, complexity and diversity are increasingly being recognized."

Line 76 – what is an "underlying environmental parameter"…? Try to use plain language.

Lines 81-88 – It seems odd to analyse ASPAs in regions of Antarctica in terms of the proportion of the species in a region that occur in the ASPA in that region.. Why not focus on which ASPAs are providing the greatest representation of species in Antarctica – ie which ASPAs are best performing in terms of biodiversity representation. If an ASPA includes only 8% of the species in a region, but that region contains 90% of the species in the continent, then that might be regarded as a really top performing ASPA. I guess my concern with this passage is that its intent is unclear. What is the significance of these numbers? Perhaps provide that context please?

Paragraph starting line 90 – This passage highlights the problem of an ASPA-based analysis. And starts to provide some of the important context that I think needs to come sooner. What is the benefit of an ASPA beyond being outside an ASPA? Is there a more rigorous impact assessment process? Are some things just not permitted in ASPAs that are permitted elsewhere?

The figure needs to show how biodiversity (species richness) is geographically distributed across the continent for the % figures to make any sense

REFERENCES

Wilson et al. *Biological Conservation* (2005) 22 (1): 99-112.

<http://dx.doi.org/10.1016/j.biocon.2004.07.004>

Rondinini, C. , Wilson, K. A., Boitani, L. , Grantham, H. and Possingham, H. P. (2006), Tradeoffs of

different types of species occurrence data for use in systematic conservation planning. *Ecology Letters*, 9: 1136-1145. doi:10.1111/j.1461-0248.2006.00970.x

A snapshot of biodiversity protection in Antarctica

Wauchope, H., Shaw, J.D. & Terauds, A.*

*Corresponding author

RESPONSE TO REVIEWERS

Reviewers' comments:

Reviewer #1 (Remarks to the Author):

A snapshot of biodiversity protection in Antarctica

Wauchope et al.

Reviewed by Kevin A. Hughes, British Antarctic Survey, Cambridge, UK.

General comments

The authors have tackled a question that is one of the major factors holding back progress with the development of the Antarctic Protected Area System through the Antarctic Treaty System's Committee for Environmental Protection. The conclusions represent a significant and major advance in the field that will be of great benefit to scientist and policy makers within the 40 nations that have signed the Protocol on Environmental Protection to the Antarctic Treaty.

RESPONSE: We appreciate the acknowledgement that this work represents a significant and major advance, with high potential to drive policy change.

Annex V to the Protocol, which describes the mechanism and criteria for area protection, is rather vague and which has led to inconsistencies in the development of the ASPA network. Added to this, the majority of ASPAs were designated before Annex V entered into force, previously being SSSIs or SPAs. This has led to a very confused system which makes analysis difficult, and inevitably the authors have had to make some judgement calls during this research which may be open for debate. Nevertheless, Wauchope et al. have used a sensible and carefully considered methodology that has moved the whole areas forward very significantly. I have made a few comments below, suggesting areas where their analysis might be re-examined, but overall I see this work as substantially progressing the field and I look forward to seeing it published.

Specific comments

Page 1 Line 18: It might be helpful to briefly mention which threats to Antarctic biodiversity are escalating.

RESPONSE: We have now added text to the Abstract detailing the threats

Page 1. Line 30-31. Antarctic had protection before 1998 through the Agreed Measures for Conservation of Antarctic Flora and Fauna (1964). Much of the Protocol built on the Agreed Measures. Also, Annex V did not enter into force until 2002 (not 1998).

RESPONSE: We thank the reviewer for this very helpful feedback. We have now included a reference to the Agreed Measures (L 33-34) and also clarified the date that Annex V came into force the first time it is mentioned in the text (L44)

Page 2 Line 65. Semi-colon before 'however'

Added as suggested

Page 3 Line 67 and Methods Page 7 Line 207-211. I wonder if the authors can explain in more detail their rationale for excluding seals and emperor penguins from the study, particularly, for example, as ASPA 137 Northwest White Island was designated with the primary object of protecting the Weddell Seal colony located there and several ASPA (e.g. APSAs 101, 107, 127, etc.) have been designated primarily to protect emperor penguins. The well-documented breeding locations of these species on ice, might mean that their inclusion is possible.

RESPONSE: While we had initially included some emperor penguin locations in our database, these were excluded from the analyses based on what we saw as insufficient data coverage. In response to this suggestion we have expanded our search effort and found a comprehensive source of emperor penguin records (from the MAPPD database <http://www.penguinmap.com/Species/>) and have now also included them in our database. The MAPPD database also contained comprehensive records on Chinstrap and Gentoo Penguins and these have also been included. We have re-run all analyses with these new data and updated the results and figures. We have also amended the Methods to reflect the inclusion of emperor penguins (L281 – 285).

In regard to Weddell seals - throughout the data compilation process, records of Weddell seals were not included as our primary focus was on compiling a database of non-marine species. As Weddell seals breed on ice and are a marine species, we did not collect data on them and therefore they are inadequately represented in our dataset. Furthermore, unlike Emperor penguins, Weddell seals are much more disparate in their continent wide distribution and there are no comprehensive data sources on their breeding locations at this stage. We therefore feel it would be biased to include them in our analysis.

Page 3 Line 69-72. I found it interesting that fungi were included in the analysis. I am not sure that the data from more recent molecular biodiversity work is routinely incorporated into the Antarctic databases used for this analysis. Vascular plants, bryophytes and lichens have probably been characterised to the greatest extent. However, existing fungal diversity records often rely upon earlier culture work and may not take into account the much wider diversity detected by, for example, environmental DNA methodologies. I wonder how confident the authors are that this analysis of fungi is fully representative of the available data.

RESPONSE: We initially used the term 'plants and fungi' to encapsulate the fungal component of lichens. We acknowledge that the taxonomy of microbes is problematic, and for this reason we have now removed all data on microbes (primarily bacteria, chromista and protozoa) from our analyses and we have also clarified this in the Methods (L288 - 289). To ensure that our analyses were not biased by the issues raised by this reviewer, we also refined our categorisation of plants and fungi, to ensure that we only included fungi that were associated with lichens (see Methods – L295-296) and we now refer to that group, more correctly, as plants and lichens throughout the manuscript. We re-ran the analyses with these revised data, which has resulted to some minor

modifications to the Results and some of the figures, but not altered our overall conclusions.

Page 3 Line 86-88. The size, rather than the number of ASPAs in a particular ACBR may put a different perspective on the results shown (i.e. see Figure 1d and Supplementary Figure 1a and c). For example, the limited number of ASPAs in ACBR 4 are large in size compared to most ASPAs, and are also the location where much of the biodiversity research has been done. This may explain the high degree of species known from the ACBR that are found within ASPAs. However, other areas within this ACBR have been little explored and the overall biodiversity of the ACBRs is likely be greater. The authors have described some of these issues on page 4 lines 121 onward, but perhaps some of the complexities could be at least mentioned earlier in the manuscript, to give context to the examples provided in this example.

RESPONSE: This is an interesting point and we agree that an earlier emphasis on these issues would be useful. Therefore, we have now mentioned it earlier in the ms (L123 - 126). We also now include a number of measures to investigate issues associated with the area of ASPAs compared to the number of ASPAs, including two new Supplementary Figures that explicitly consider area. Supplementary Figure 2 shows the number of species in each ACBR per km² of ASPA that they occur in, and Supplementary Figure 3 is a modified version of Fig 1d that shows the 10 most protected species according to area, instead of number of ASPAs. We also refer to these results in the main body of the text (L179 - 182); however, we have chosen to keep 'number of ASPAs' as our primary metric rather than area (ie in Fig. 1) as the 'area' figures are heavily biased by several large ASPAs (e.g. 126, 147, 170, 173). Point records within these ASPAs or data from the management plans do not generally give an indication of the distribution of the species within the ASPA, and therefore, in our view, it is could be misleading to suggest that a species is receiving 100s of km² of ASPA protection when it's actual range is unknown. We now also include a brief discussion of this in the caption of Supplementary Figure 4. To further address the points raised by this reviewer, we have expanded on the point that ASPAs have different levels of survey effort (L86-87) within a broad paragraph that covers several of the data limitation issues (L83-96)

Page 3 Line 99 to Page 4 Line 107 and Supplementary Table 2. Looking at the list in Supplementary Table 2, I think the author may have been a little simplistic with their allocation of 'Protocol designation', and not perhaps adequately recognised that multiple values can be protected within an ASPA. The authors could consider re-labelling this column 'Primary Protocol Designation'. This also relates to how the information is shown in Figure 1a. I'm not sure this does justice to ASPAs protecting a combination of values.

RESPONSE: As this reviewer suggests, we have amended the column to state 'primary protocol designation', and have also clarified this in the caption of Figure 1c. We have also expanded the text around the issues of multiple values occurring within ASPAs (L139-153). However, in our view, even though multiple values may be mentioned in ASPA management plans, it is rarely clear if the sites are managed for multiple values and we have also tried to cover this in more detail in this expanded paragraph. In our experience, ASPAs are most often managed for their primary protocol designation, even if other values are present and one of the key points from this paper is that in any systematic development of area protection in Antarctica, ASPAs should protect multiple values, and explicitly provide measures for this protection in the Management Plans (see L160-165 and L202-205).

The authors could also add a column detailing in which ACBR the ASPA is found.

RESPONSE: This column has now been added.

I note that the authors make little mention of the ASPAs that protect predominantly marine biodiversity. It might be useful to explain a little more how they fit (or not) into this study. Is this an opportunity for further work?

RESPONSE: We have now added text (and a supporting reference) to address this issue, noting that we did not include these predominantly marine ASPAs due to a lack of available data (both more broadly and from ASPAs) and our focus on terrestrial protection, but that it does present an opportunity for future research (L184-190).

I would also suggest that the authors check a few of the ASPAs that have been described as not being designated for protection of biodiversity (with '0' in column 4). I think that ecological values have been included in some of these ASPA. I have provided some text from the relevant ASPA Management Plans for some examples, which show that biological values are part of the reason for designation:

- ASPA 123 Barwick and Balham Valleys 'Some of the best examples of the physical surface features associated with this unique and extreme environment are found on the valley floors, where there are also fine examples of microbial life, lichens, as well as soil and lake microflora.'
- ASPA 172. Lower Taylor Glacier and Blood Falls. The primary reasons for designation of the Area are its unique physical properties, and the unusual microbial ecology and geochemistry.
- ASPA 147. Ablation Valley and Ganymede Heights. The primary reason for the designation of Ablation Valley and Ganymede Heights, Alexander Island (70°48'S, 68°30'W, approximately 180 km²) as an Antarctic Specially Protected Area (ASPAs) is to protect scientific values, relating particularly to the geology, geomorphology, glaciology, limnology and ecology of this extensive ablation area.

RESPONSE: We thank the reviewer for pointing out this text in these Management Plans. We have now included ASPA 123 in our category of biodiversity ASPAs. However, after careful consideration, we have not included ASPA 172 (as microbial life was not included in our analyses), or ASPA 147, as it was designated to protect 'scientific values' which does not necessarily offer the same protection as the other designations (see also our text and citations on this issue L155 - 165). We acknowledge that the issue of protecting microbial communities is of increasing interest in Antarctica and we have now included some text (with supporting citations) on this (L190-194)

Following on from our response above, in our view it is not always clear if an ASPA is being managed for multiple values, and in our experience, the primary designation reason usually receives the most management focus. As we indicated in our above response, in any systematic improvement of area protection, ASPAs would ideally be managed for multiple values where possible, and that this should be explicitly addressed in the Management Plans, and we have tried to emphasise this point in our revised text (see L160-165 and L202-205).

Page 4 Line 109-111. I note the emphasis has been placed on the presence of species in the greatest 'number' of ASPAs (see earlier comment). Rather I wonder if it is possible to take ASPA areas into

consideration. For example, ASPA 126 Byers Peninsula has an area of 90 km² and ASPA 147 Ablation Valley has an area of 110 km². These values contrast greatly with the much smaller areas of many other ASPAs, such as ASPA 110 Lynch Island with an area of only 0.15 km². Hence some mention of ASPA area would be important and give more accurate representation of the relative level of protection of some species.

RESPONSE: We have now taken area into account in extra analyses (see above text), and this is reflected in both the text and the new Supplementary Figures.

Page 4 Line 130. Change to 'Annex V to the Protocol'

RESPONSE: Changed as suggested

Page 5 Line 136-137. Here the authors write 'Here we provide an important spatial- and species specific assessment to inform and assist targeted continent-wide systematic area protection for terrestrial Antarctica.' I agree fully with this assessment and would suggest that this stated earlier in the manuscript.

RESPONSE: Now paraphrased and included earlier in the manuscript (L89 – 96)

Page 5 Line 141-146. Not necessarily for inclusion here, but the authors may like to consider the work of Hughes and Grant which is currently in press in Polar Research entitled: 'Current logistical capacity is sufficient to deliver the implementation and management of a representative Antarctic protected area system'

RESPONSE: We thank the reviewer for bringing this to our attention

Page 5 Line 162. I think there are 72 ASPAs, not 73, i.e. number 101 to 175 (75), minus ASPAs 114, 118 and 130 (3) gives 72 in total.

RESPONSE: We have now corrected this error

Page 6 Lines 180-201. The work to resolve the boundary issues is of great value as this has been an issue that had cause difficulties for scientist and policy makers in the past. The approach taken seems reasonable, given the complexities.

RESPONSE: We thank the reviewer for acknowledging the value of revising this spatial layer, it represents a considerable amount of work.

Page 7. Lines 226-228. The authors should recheck some of the ASPAs that have been described as 'non-biodiversity designated ASPAs', e.g. ASPAs 147, 123, and possibly amend the size of the 'purple' bars as appropriate in Supplementary Figures 1b and 1d. This may also lead to tweaks of Supplementary Figure 2 and part so Figure 1.

RESPONSE: We thank the reviewer for pointing this out. Please see our above response regarding primary protocol designation.

References.

The references listed seem reasonable and appropriate for this research area.

RESPONSE: Some extra references have now been added to support the changes made in response to suggestions from both reviewers.

Supplementary Table 1. The Environmental Protocol in Annex V states: 'Any area, including any marine area, may be designated as an Antarctic Specially Protected Area to protect outstanding environmental, scientific, historic, aesthetic or wilderness values, any combination of those values, or ongoing or planned scientific research.' I think it is important that this broader concept of multiple values being protected in a single ASPA is highlighted, particularly in light of the statements made on Page 4 Line 104-107, i.e. 'These findings highlight the opportunity for proponents of ASPAs to consider managing for multiple objectives, ensuring appropriate management for the original designation, the associated biodiversity, and any other values that are present.'

RESPONSE: We have now added extra text to expand on this issue earlier in the manuscript (L139-153)

I note that this work is supported by the SCAR Standing Committee on the Antarctic Treaty System, which is the conduit for communication between SCAR and the Antarctic Treaty System, including the Committee for Environmental Protection, which oversees ASPA management plan preparation. Based on the comments here, I wonder if SCAR would be will to provide biodiversity data, revealed in the research, for new ASPAs, or during the revision of existing ASPA management plans.

RESPONSE: The biodiversity data that we extracted from the ASPA Management Plans will be made publicly available on publication (data.aad.gov.au – DOI pending). The authors are working closely with SCAR and a number of researchers in the Antarctic community to make the entire terrestrial biodiversity database publicly available through a published data paper, due for submission in early 2019.

Reviewer #2 (Remarks to the Author):

Overall

I find this manuscript to be well written and the analysis methods simple and easy to understand. I was surprised that the authors don't take the opportunity to provide specific advice about where the next best places might be to place protected areas.

RESPONSE: The Antarctic Treaty system's Committee for Environmental Protection has committed to developing protected areas in a systematic way, including the assessment of a range of spatial layers, values and deficiencies. The work presented here represents an important foundational element of this process, but it is one of a range of potentially diverse inputs. In this work we highlight several areas that need more attention in any further protected area development, including addressing taxonomic and region specific biases and the management of multiple values. This work has been specifically designed to maximise impact in this international policy arena and our Results and conclusions are aligned with the requirements of the Protocol on Environmental Protection to the Antarctic Treaty. In this context we believe spatially explicit recommendations for new protected areas are beyond the scope of this particular paper, and better addressed in future work that considers a more holistic range of inputs.

I was also surprised to see the authors use only point records of biodiversity as a means of assessing the ASPAs given that this method of evaluating and designing conservation reserves appears to have

some significant limitations compared with using modelled distributions (Wilson et al. 2004; Rondinini et al. 2006). This strategy needs to be clearly justified against alternatives, including utilising modelled species distributions. Clearly, there could be some species present in ASPAs that have not been observed there yet, and there could be some species present in eco-regions that have not been recorded in those regions yet. This needs to be at least acknowledged, if not explicitly dealt with in analysis.

RESPONSE: We agree with this reviewer that there is considerable potential for species distribution models to build and expand on the work presented here. While we acknowledge that there are limitations with the use of point data from surveys for these type of assessments, we also note that there are considerable complexities and limitations associated with using modelled data of the type suggested by the reviewer, especially when attempting to combine the predicted distributions of multiple taxa that may be based on disparate environmental data of varying quality and resolution. In response, we have now included a paragraph justifying our use of occurrence data and acknowledging the potential utility of modelled data (L83 – L96).

Generally, the point/value of ASPAs was not particularly well introduced. Given that Antarctica has “had unique continent-wide protection since 1998, under the Protocol on Environmental Protection to the Antarctic Treaty “, it is hard to feel strongly motivated by the representativeness or otherwise of the ASPAs.. Especially when it appears that a bunch of them are not actually aimed at protecting biodiversity and even when they are, it is not clear what actually happens on them that doesn’t happen elsewhere (or vice-versa). How many of the species in your analysis are faced with a higher risk of extinction because they’re not in ASPAs?

RESPONSE: We thank the reviewer for identifying this deficiency in our original submission and we appreciate the perspective they provide on this aspect. We have now provided more detail on some of the negative impacts that are already occurring under the broader level of protection provided under the Protocol (L40-42) and included a new paragraph detailing the importance of ASPAs as a critical mechanism for protecting Antarctic biodiversity (L48-61).

We agree with the reviewer that ASPAs that are not designated for biodiversity are of questionable conservation value, and this leads us to one of our key conclusions - that in any systematic revision of area protection, multiple values need to be considered and explicitly managed (see L160-165 and L202-205). We also discuss this issue in more detail in an expanded paragraph on this issue (L139-153). Explicitly quantifying species’ extinction prospects in and out of ASPAs would be a huge body of work requiring extensive knowledge of survival requirements that is not available in most cases. Such work is beyond the scope of this paper; however, in our revised manuscript we believe we have now made a much stronger case that ASPAs are a key conservation measure for protecting biodiversity in Antarctica.

To that end, I was left feeling unclear about whether the representativeness is really a very good way to assess conservation effort/outcome on Antarctica. Some assessment of adequacy, referred to early in the manuscript would seem very appropriate, but that is not currently provided. If climate change is the top driver of future biodiversity loss in Antarctica, then how will ASPAs help?

RESPONSE: Our focus is on providing information that is aligned with the requirements of the Protocol on Environmental Protection to the Antarctic Treaty System, as experience has shown that this is the most effective way to drive policy change and achieve conservation related outcomes in the consensus driven Antarctic Treaty systems Committee for Environmental

Protection. The continental representation of biodiversity is strongly aligned with the call in the Protocol for ASPAs to be defined within a systematic environmental-geographic framework (see our text L68-72 and extra supporting references), and as acknowledged by Reviewer 1, quantifying this representation is a key element in progressing the development of systematic area protection in the international policy arena. In this context, ASPA representativeness is a fundamental measure when assessing conservation efforts and outcomes in Antarctica.

As we acknowledge in our response to the first general point from this reviewer, this work is one element in a suite of inputs that will ultimately be used to inform more systematic development of area protection in Antarctica, and we agree that further work on adequacy is also required. In addition, we have now added extra text clarifying the effectiveness of ASPAs in relation to climate change (L56- 58).

Specific comments

Line 18 – is it all of biodiversity (including insects and bryophytes?) or just vertebrates – mention here so readers can get an idea before diving into the paper.

RESPONSE: We have now added appropriate details to the Abstract text.

Lines 36-40 – Given the apparently strong protection of biodiversity on Antarctica, it is unclear what additional protection ASPAs provide. Therefore it is hard to feel strongly motivated by the need to see how representative these ASPAs are in terms of biodiversity. Can you motivate this story a bit more with something about the biodiversity impacts that are mitigated/avoided on ASPAs? (and how?)

RESPONSE: We have now reinforced some of the impacts that are already occurring under the broader level of protection provided under the Protocol (L40-42) and included a new paragraph detailing the importance of ASPAs as critical mechanism for protecting Antarctic biodiversity (L48-61).

Line 49. Vague. What discussions? Any in particular? What do you mean by area protection? Could you refer to any international agreements here?

We have now added text to specify that the main international arena for these discussions is the Antarctic Treaty systems Committee for Environmental Protection. We have also cited a recent submission to the CEP that exemplifies these discussions, and also included citations of the key scientific literature (L65-67).

Line 55 – what is an ASPA spatial planning layer? What does it contain? Why would that be interesting?

RESPONSE: We have amended the text to clarify that we were referring to the spatial polygon layer of the ASPA locations and boundaries.

Line 62 – This statement doesn't seem to fit with the rest of the paragraph. Why would the perceived 'simplicity' of Antarctic ecosystems be relevant to the rest of this paragraph which seems to be about where the biodiversity is and how well protected it is. "While Antarctic terrestrial ecosystems have often been considered relatively simple, their structure, complexity and diversity are increasingly being recognized."

RESPONSE: We have now deleted this sentence and added more relevant detail to the following sentence regarding the type of biodiversity we are introducing in this sentence (L100-102)

Line 76 – what is an “underlying environmental parameter”..? Try to use plain language.

RESPONSE: We have now amended the text to clarify (L 114)

Lines 81-88 – It seems odd to analyse ASPAs in regions of Antarctica in terms of the proportion of the species in a region that occur in the ASPA in that region.. Why not focus on which ASPAs are providing the greatest representation of species in Antarctica – ie which ASPAs are best performing in terms of biodiversity representation. If an ASPA includes only 8% of the species in a region, but that region contains 90% of the species in the continent, then that might be regarded as a really top performing ASPA. I guess my concern with this passage is that its intent is unclear. What is the significance of these numbers? Perhaps provide that context please?

RESPONSE: This is a good point and we have amended Figure 1b to improve the context of the figure and the information that it illustrates. We now display the number of protected species in an ecoregion as a percentage of both the number of species in that ecoregion and as a percentage of all Antarctic species and make reference to both sets of results in the text (L112 – 123). We also provide an extra Supplementary Figure that shows the distribution of species richness across the Antarctic continent (Supp Figure 2).

Paragraph starting line 90 – This passage highlights the problem of an ASPA-based analysis. And starts to provide some of the important context that I think needs to come sooner. What is the benefit of an ASPA beyond being outside an ASPA? Is there a more rigorous impact assessment process? Are some things just not permitted in ASPAs that are permitted elsewhere?

RESPONSE: We have now added extra text and supporting references in the introductory paragraph to highlight impacts in areas outside of ASPAs (L40-42) and also provided a new paragraph detailing the clear conservation benefits of ASPAs for biodiversity in Antarctica (L48-61)

The figure needs to show how biodiversity (species richness) is geographically distributed across the continent for the % figures to make any sense

RESPONSE: In addition to the additional context provided in Figure 1b (see above response) we have also provided a new Supplementary Figure that illustrates species richness across the continent, as the number of species protected per km² of ASPA (Supplementary Figure 2). In our view these additions now provide the necessary context requested by this reviewer. We have also referred to these Results in the body of the main ms (L112 - 123)

REFERENCES

- Wilson et al. Biological Conservation (2005) 22 (1): 99-112. <http://dx.doi.org/10.1016/j.biocon.2004.07.004>
- Rondinini, C. , Wilson, K. A., Boitani, L. , Grantham, H. and Possingham, H. P. (2006), Tradeoffs of different types of species occurrence data for use in systematic conservation planning. Ecology Letters, 9: 1136-1145. doi:10.1111/j.1461-0248.2006.00970.x

RESPONSE: We thank the reviewer for drawing our attention to these references, we have now included citations to them in the new paragraph that addresses the utility of species distribution models (L85-89).

REVIEWERS' COMMENTS:

Reviewer #1 (Remarks to the Author):

REVIEWER #1: K. A. Hughes. Please find comments on the revisions incorporated into the text below:

A snapshot of biodiversity protection in Antarctica

Wauchope, H., Shaw, J.D. & Terauds, A.*

*Corresponding author

RESPONSE TO REVIEWERS

Reviewers' comments:

Reviewer #1 (Remarks to the Author):

A snapshot of biodiversity protection in Antarctica

Wauchope et al.

Reviewed by Kevin A. Hughes, British Antarctic Survey, Cambridge, UK.

General comments

The authors have tackled a question that is one of the major factors holding back progress with the development of the Antarctic Protected Area System through the Antarctic Treaty System's Committee for Environmental Protection. The conclusions represent a significant and major advance in the field that will be of great benefit to scientist and policy makers within the 40 nations that have signed the Protocol on Environmental Protection to the Antarctic Treaty.

RESPONSE: We appreciate the acknowledgement that this work represents a significant and major advance, with high potential to drive policy change.

Annex V to the Protocol, which describes the mechanism and criteria for area protection, is rather vague and which has led to inconsistencies in the development of the ASPA network. Added to this, the majority of ASPAs were designated before Annex V entered into force, previously being SSSIs or SPAs. This has led to a very confused system which makes analysis difficult, and inevitably the authors have had to make some judgement calls during this research which may be open for debate. Nevertheless, Wauchope et al. have used a sensible and carefully considered methodology that has moved the whole areas forward very significantly. I have made a few comments below, suggesting areas where their analysis might be re-examined, but overall I see this work as substantially progressing the field and I look forward to seeing it published.

Specific comments

Page 1 Line 18: It might be helpful to briefly mention which threats to Antarctic biodiversity are escalating.

RESPONSE: We have now added text to the Abstract detailing the threats

REVIEWER #1: Thank you

Page 1. Line 30-31. Antarctic had protection before 1998 through the Agreed Measures for Conservation of Antarctic Flora and Fauna (1964). Much of the Protocol built on the Agreed Measures. Also, Annex V did not enter into force until 2002 (not 1998).

RESPONSE: We thank the reviewer for this very helpful feedback. We have now included a reference to the Agreed Measures (L 33-34) and also clarified the date that Annex V came into force the first time it is mentioned in the text (L44)

REVIEWER #1: Thank you

Page 2 Line 65. Semi-colon before 'however'

Added as suggested

REVIEWER #1: Thank you

Page 3 Line 67 and Methods Page 7 Line 207-211. I wonder if the authors can explain in more detail their rationale for excluding seals and emperor penguins from the study, particularly, for example, as ASPA 137 Northwest White Island was designated with the primary object of protecting the Weddell Seal colony located there and several ASPA (e.g. APSAs 101, 107, 127, etc.) have been designated primarily to protect emperor penguins. The well-documented breeding locations of these species on ice, might mean that their inclusion is possible.

RESPONSE: While we had initially included some emperor penguin locations in our database, these were excluded from the analyses based on what we saw as insufficient data coverage. In response to this suggestion we have expanded our search effort and found a comprehensive source of emperor penguin records (from the MAPPD database <http://www.penguinmap.com/Species/>) and have now also included them in our database. The MAPPD database also contained comprehensive records on Chinstrap and Gentoo Penguins and these have also been included. We have re-run all analyses with these new data and updated the results and figures. We have also amended the Methods to reflect the inclusion of emperor penguins (L281 – 285).

REVIEWER #1: Thank you

In regard to Weddell seals - throughout the data compilation process, records of Weddell seals were not included as our primary focus was on compiling a database of non-marine species. As Weddell seals breed on ice and are a marine species, we did not collect data on them and therefore they are inadequately represented in our dataset. Furthermore, unlike Emperor penguins, Weddell seals are much more disparate in their continent wide distribution and there are no comprehensive data sources on their breeding locations at this stage. We therefore feel it would be biased to include them in our analysis.

REVIEWER #1: I agree with the authors that there are significant difficulties associated with incorporating seals into their analysis, which could result in potentially misleading conclusions. Given that investigation of predominantly marine species was not the main aim of the work I think this response is reasonable.

Page 3 Line 69-72. I found it interesting that fungi were included in the analysis. I am not sure that the data from more recent molecular biodiversity work is routinely incorporated into the Antarctic databases used for this analysis. Vascular plants, bryophytes and lichens have probably been characterised to the greatest extent. However, existing fungal diversity records often rely upon earlier culture work and may not take into account the much wider diversity detected by, for example, environmental DNA methodologies. I wonder how confident the authors are that this analysis of fungi is fully representative of the available data.

RESPONSE: We initially used the term 'plants and fungi' to encapsulate the fungal component of lichens. We acknowledge that the taxonomy of microbes is problematic, and for this reason we have now removed all data on microbes (primarily bacteria, chromista and protozoa) from our analyses and we have also clarified this in the Methods (L288 - 289). To ensure that our analyses were not biased by the issues raised by this reviewer, we also refined our categorisation of plants and fungi, to ensure that we only included fungi that were associated with lichens (see Methods – L295 296) and we now refer to that group, more correctly, as plants and lichens throughout the manuscript. We re-ran the analyses with these revised data, which has resulted to some minor modifications to the Results and some of the figures, but not altered our overall conclusions.

REVIEWER #1: Thank you for clarifying your categorisation and reconsidering how fungi were incorporated into your analysis.

Page 3 Line 86-88. The size, rather than the number of ASPAs in a particular ACBR may put a different perspective on the results shown (i.e. see Figure 1d and Supplementary Figure 1a and c). For example, the limited number of ASPAs in ACBR 4 are large in size compared to most ASPAs,

and are also the location where much of the biodiversity research has been done. This may explain the high degree of species known from the ACBR that are found within ASPAs. However, other areas within this ACBR have been little explored and the overall biodiversity of the ACBRs is likely be greater. The authors have described some of these issues on page 4 lines 121 onward, but perhaps some of the complexities could be at least mentioned earlier in the manuscript, to give context to the examples provided in this example.

RESPONSE: This is an interesting point and we agree that an earlier emphasis on these issues would be useful. Therefore, we have now mentioned it earlier in the ms (L123 - 126). We also now include a number of measures to investigate issues associated with the area of ASPAs compared to the number of ASPAs, including two new Supplementary Figures that explicitly consider area. Supplementary Figure 2 shows the number of species in each ACBR per km² of ASPA that they occur in, and Supplementary Figure 3 is a modified version of Fig 1d that shows the 10 most protected species according to area, instead of number of ASPAs. We also refer to these results in the main body of the text (L179 - 182); however, we have chosen to keep 'number of ASPAs' as our primary metric rather than area (ie in Fig. 1) as the 'area' figures are heavily biased by several large ASPAs (e.g. 126, 147, 170, 173). Point records within these ASPAs or data from the management plans do not generally give an indication of the distribution of the species within the ASPA, and therefore, in our view, it is could be misleading to suggest that a species is receiving 100s of km² of ASPA protection when it's actual range is unknown. We now also include a brief discussion of this in the caption of Supplementary Figure 4. To further address the points raised by this reviewer, we have expanded on the point that ASPAs have different levels of survey effort (L86-87) within a broad paragraph that covers several of the data limitation issues (L83-96)

REVIEWER #1: The two new figures considering area are useful – thank you. The authors make a reasonable point concerning the distribution of a particular species within an ASPAs, so I would support their use of 'number of ASPAs' as the primary metric, given the data available.

Page 3 Line 99 to Page 4 Line 107 and Supplementary Table 2. Looking at the list in Supplementary Table 2, I think the author may have been a little simplistic with their allocation of 'Protocol designation', and not perhaps adequately recognised that multiple values can be protected within an ASPA. The authors could consider re-labelling this column 'Primary Protocol Designation'. This also relates to how the information is shown in Figure 1a. I'm not sure this does justice to ASPAs protecting a combination of values.

RESPONSE: As this reviewer suggests, we have amended the column to state 'primary protocol designation', and have also clarified this in the caption of Figure 1c. We have also expanded the text around the issues of multiple values occurring within ASPAs (L139-153). However, in our view, even though multiple values may be mentioned in ASPA management plans, it is rarely clear if the sites are managed for multiple values and we have also tried to cover this in more detail in this expanded paragraph. In our experience, ASPAs are most often managed for their primary protocol designation, even if other values are present and one of the key points from this paper is that in any systematic development of area protection in Antarctica, ASPAs should protect multiple values, and explicitly provide measures for this protection in the Management Plans (see L160-165 and L202-205).

REVIEWER #1: The expansion of the text around the issues of multiple values occurring within ASPAs is particularly useful and further strengthens the manuscript. I would agree that ASPA management activities often focus on the primary value under protection. How individual ASPAs contribute to continent-wide protection of biodiversity is something that is considered rarely by environmental managers responsible for individual ASPAs, which is a factor that could be usefully brought to the attention of the Antarctic Treaty Consultative Meeting's Committee for Environmental Protection, which oversees ASPA management plan drafting and revision.

The authors could also add a column detailing in which ACBR the ASPA is found.

RESPONSE: This column has now been added.

REVIEWER #1: Thank you.

I note that the authors make little mention of the ASPAs that protect predominantly marine

biodiversity. It might be useful to explain a little more how they fit (or not) into this study. Is this an opportunity for further work?

RESPONSE: We have now added text (and a supporting reference) to address this issue, noting that we did not include these predominantly marine ASPAs due to a lack of available data (both more broadly and from ASPAs) and our focus on terrestrial protection, but that it does present an opportunity for future research (L184-190).

REVIEWER #1: Thank you.

I would also suggest that the authors check a few of the ASPAs that have been described as not being designated for protection of biodiversity (with 'O' in column 4). I think that ecological values have been included in some of these ASPA. I have provided some text from the relevant ASPA Management Plans for some examples, which show that biological values are part of the reason for designation:

- ASPA 123 Barwick and Balham Valleys ' Some of the best examples of the physical surface features associated with this unique and extreme environment are found on the valley floors, where there are also fine examples of microbial life, lichens, as well as soil and lake microflora.'
- ASPA 172. Lower Taylor Glacier and Blood Falls. The primary reasons for designation of the Area are its unique physical properties, and the unusual microbial ecology and geochemistry.
- ASPA 147. Ablation Valley and Ganymede Heights. The primary reason for the designation of Ablation Valley and Ganymede Heights, Alexander Island (70°48'S, 68°30'W, approximately 180 km²) as an Antarctic Specially Protected Area (ASPAs) is to protect scientific values, relating particularly to the geology, geomorphology, glaciology, limnology and ecology of this extensive ablation area.

RESPONSE: We thank the reviewer for pointing out this text in these Management Plans. We have now included ASPA 123 in our category of biodiversity ASPAs. However, after careful consideration, we have not included ASPA 172 (as microbial life was not included in our analyses), or ASPA 147, as it was designated to protect 'scientific values' which does not necessarily offer the same protection as the other designations (see also our text and citations on this issue L155 - 165). We acknowledge that the issue of protecting microbial communities is of increasing interest in Antarctica and we have now included some text (with supporting citations) on this (L190-194). Following on from our response above, in our view it is not always clear if an ASPA is being managed for multiple values, and in our experience, the primary designation reason usually receives the most management focus. As we indicated in our above response, in any systematic improvement of area protection, ASPAs would ideally be managed for multiple values where possible, and that this should be explicitly addressed in the Management Plans, and we have tried to emphasise this point in our revised text (see L160-165 and L202-205).

REVIEWER #1: Thank you for taking the recommended amendments into consideration and for justifying the inclusion, or otherwise, of the further ASPAs within your analysis. The justifications for excluding ASPA 172 seems reasonable. With regard to ASPA 147, it could be useful for the Committee for Environmental Protection to clarify how protection of biodiversity might differ (or not!) when protecting an area for 'scientific' or 'environmental values'. However, it is not necessarily appropriate to go into that debate within this paper.

Page 4 Line 109-111. I note the emphasis has been placed on the presence of species in the greatest 'number' of ASPAs (see earlier comment). Rather I wonder if it is possible to take ASPA areas into consideration. For example, ASPA 126 Byers Peninsula has an area of 90 km² and ASPA 147 Ablation Valley has an area of 110 km². These values contrast greatly with the much smaller areas of many other ASPAs, such as ASPA 110 Lynch Island with an area of only 0.15 km². Hence some mention of ASPA area would be important and give more accurate representation of the relative level of protection of some species.

RESPONSE: We have now taken area into account in extra analyses (see above text), and this is

reflected in both the text and the new Supplementary Figures.

REVIEWER #1: Thank you.

Page 4 Line 130. Change to 'Annex V to the Protocol'

RESPONSE: Changed as suggested

REVIEWER #1: Thank you.

Page 5 Line 136-137. Here the authors write 'Here we provide an important spatial- and species specific assessment to inform and assist targeted continent-wide systematic area protection for terrestrial Antarctica.' I agree fully with this assessment and would suggest that this stated earlier in the manuscript.

RESPONSE: Now paraphrased and included earlier in the manuscript (L89 – 96)

REVIEWER #1: Thank you.

Page 5 Line 141-146. Not necessarily for inclusion here, but the authors may like to consider the work of Hughes and Grant which is currently in press in Polar Research entitled: 'Current logistical capacity is sufficient to deliver the implementation and management of a representative Antarctic protected area system'

RESPONSE: We thank the reviewer for bringing this to our attention

Page 5 Line 162. I think there are 72 ASPAs, not 73, i.e. number 101 to 175 (75), minus ASPAs 114, 118 and 130 (3) gives 72 in total.

RESPONSE: We have now corrected this error

REVIEWER #1: Thank you.

Page 6 Lines 180-201. The work to resolve the boundary issues is of great value as this has been an issue that had cause difficulties for scientist and policy makers in the past. The approach taken seems reasonable, given the complexities.

RESPONSE: We thank the reviewer for acknowledging the value of revising this spatial layer, it represents a considerable amount of work.

Page 7. Lines 226-228. The authors should recheck some of the ASPAs that have been described as 'non-biodiversity designated ASPAs', e.g. ASPAs 147, 123, and possibly amend the size of the 'purple' bars as appropriate in Supplementary Figures 1b and 1d. This may also lead to tweaks of Supplementary Figure 2 and part so Figure 1.

RESPONSE: We thank the reviewer for pointing this out. Please see our above response regarding primary protocol designation.

REVIEWER #1: Noted, thank you.

References.

The references listed seem reasonable and appropriate for this research area.

RESPONSE: Some extra references have now been added to support the changes made in response to suggestions from both reviewers.

REVIEWER #1: The additional references seem appropriate.

Supplementary Table 1. The Environmental Protocol in Annex V states: 'Any area, including any marine area, may be designated as an Antarctic Specially Protected Area to protect outstanding environmental, scientific, historic, aesthetic or wilderness values, any combination of those values, or ongoing or planned scientific research.' I think it is important that this broader concept of multiple values being protected in a single ASPA is highlighted, particularly in light of the statements made on Page 4 Line 104-107, i.e. 'These findings highlight the opportunity for proponents of ASPAs to consider managing for multiple objectives, ensuring appropriate management for the original designation, the associated biodiversity, and any other values that are present.'

RESPONSE: We have now added extra text to expand on this issue earlier in the manuscript

(L139-153)

REVIEWER #1: This further text is very useful, thank you.

I note that this work is supported by the SCAR Standing Committee on the Antarctic Treaty System, which is the conduit for communication between SCAR and the Antarctic Treaty System, including the Committee for Environmental Protection, which oversees ASPA management plan preparation. Based on the comments here, I wonder if SCAR would be will to provide biodiversity data, revealed in the research, for new ASPAs, or during the revision of existing ASPA management plans.

RESPONSE: The biodiversity data that we extracted from the ASPA Management Plans will be made publicly available on publication (data.aad.gov.au – DOI pending). The authors are working closely with SCAR and a number of researchers in the Antarctic community to make the entire terrestrial biodiversity database publicly available through a published data paper, due for submission in early 2019.

REVIEWER #1: The availability of the data will be a major benefit to Antarctic environmental managers and could be a useful source of information for inclusion in new and existing ASPA (and indeed ASMA (Antarctic Specially Managed Area)) management plans.

REVIEWER #1: Additional comments (minor)

Line 228: Please close the bracket

Line 288: Are Bacteria, Chromista and Protozoa classified as kingdoms?

Line 335: 'is' available

Figure 1: In the bottom right hand corner - delete 'system'.

Supplementary Table 2. Perhaps the column headings could be clearer or spaced better.

Reviewer #2 (Remarks to the Author):

I'm happy with the text revision made by the authors about the significance and importance of ASPAs. However, I remain unconvinced by the use of only point/primary data to underpin the representativeness study. I'm sorry to sound pedantic, but I must comment on this as a scientist, not as a policy person. By not systematically analysing potential bias in their data and checking whether the findings of the study would differ substantially to findings based on an SDM analysis, I believe that the study lacks the rigour it could have. Making it less suitable for a top journal. Unless there is a really comprehensive continent-wide biodiversity data collection, the point data could be quite misleading in an analysis of the comprehensiveness of the reserve system. I'm not saying it is, but it may be, and so a comparison with SDMs, that allow you to look at what may be happening in less well-sampled parts of the continent, seems warranted.

While the authors now acknowledge the potential role of SDMs in helping to address bias and incomplete coverage in point data when assessing the adequacy of a reserve, they simply shrug and say that it's complex and beyond the scope of the study. I find this unconvincing. In arguing that combining SDMs is too complex, they cite a study by Moirano et al. that actually uses SDMs to evaluate the coverage/comprehensiveness of the Natura2000 network. The other study cited discusses the role of SDMs in community/assemblage studies which is not really relevant to this study. I'm certainly not suggesting you should do a community or assemblage model. There are now several articles in which SDMs are used to evaluate a reserve network. Here is one: <http://ec.europa.eu/environment/nature/knowledge/pdf/alterra-report-2730b.pdf>
A quick web search reveals other similar studies. All of those authors overcame the technical challenges of combining SDMs using either simple richness or complementarity analyses (e.g. Marxan/Zonation).

The reason I think this is relevant and important is because if ASPAs were designated based in part on biological surveys, using those same data to evaluate the performance of ASPAs is circular. Even worse, ASPAs may be the subject of more surveys because they are ASPAs. Whatever the reason, bias in biological survey data could be bringing optimism into your analysis of ASPA biodiversity coverage. SDMs won't solve this completely, but they do go part way by extrapolating your data beyond the survey points. The case for this is well made in the literature. To be more specific, an analysis based on SDMs would at least allow you to see whether there are other parts of the island likely to contain the values that you claim to be limited to ASPAs.

Adding this aspect to the study does not mean that the SDMs analysis would replace or over-ride the current analysis based on point data – especially if the results of both approaches were qualitatively similar. It could be put into Supp material to assure readers that the impact of possible biases in the point data have been minimized, and that the qualitative conclusions arising from the point data align with outcomes of a state-of-art SDM analysis. Note that SDM analysis need not be used for all species, it is common to combine point data and SDMs in complementarity analyses for species that do not produce good SDMs.

I must apologise for recommending this extra work (which the Editor may choose to ignore), but I believe that if you want this work to have high policy impact, you should make sure it is as rigorous as it can reasonably be given the data at hand.

These comments are not made to try to underplay the importance of the work you have done already, or the relevance of your study to policy, but simply to add the level of rigour I see as necessary to your science and policy objective.

Wauchope et al. A snapshot of biodiversity protection in Antarctica

Reviewers' comments:

Reviewer #1 (Remarks to the Author):

Reviewed by Kevin A. Hughes, British Antarctic Survey, Cambridge, UK.

REVIEWER #1: Additional comments (minor)

Line 228: Please close the bracket

Changed as suggested

Line 288: Are Bacteria, Chromista and Protozoa classified as kingdoms?

Yes, according to the Catalogue of Life, which was our key reference database for taxonomic checks and reconciliations, these are Kingdoms.

Line 335: 'is' available

Changed as suggested

Figure 1: In the bottom right hand corner - delete 'system'.

Deleted as suggested

Supplementary Table 2. Perhaps the column headings could be clearer or spaced better.

We have left-justified the headings in this table to improve readability

Reviewer #2 (Remarks to the Author):

I'm happy with the text revision made by the authors about the significance and importance of ASPAs. However, I remain unconvinced by the use of only point/primary data to underpin the representativeness study. I'm sorry to sound pedantic, but I must comment on this as a scientist, not as a policy person. By not systematically analysing potential bias in their data and checking whether the findings of the study would differ substantially to findings based on an SDM analysis, I believe that the study lacks the rigour it could have. Making it less suitable for a top journal. Unless there is a really comprehensive continent-wide biodiversity data collection, the point data could be quite misleading in an analysis of the comprehensiveness of the reserve system. I'm not saying it is, but it may be, and so a comparison with SDMs, that allow you to look at what may be happening in less well-sampled parts of the continent, seems warranted.

While the authors now acknowledge the potential role of SDMs in helping to address bias

and incomplete coverage in point data when assessing the adequacy of a reserve, they simply shrug and say that it's complex and beyond the scope of the study. I find this unconvincing. In arguing that combining SDMs is too complex, they cite a study by Moirano et al. that actually uses SDMs to evaluate the coverage/comprehensiveness of the Natura2000 network. The other study cited discusses the role of SDMs in community/assemblage studies which is not really relevant to this study. I'm certainly not suggesting you should do a community or assemblage model. There are now several articles in which SDMs are used to evaluate a reserve network. Here is one: <http://ec.europa.eu/environment/nature/knowledge/pdf/alterra-report-2730b.pdf>

A quick web search reveals other similar studies. All of those authors overcame the technical challenges of combining SDMs using either simple richness or complementarity analyses (e.g. Marxan/Zonation).

The reason I think this is relevant and important is because if ASPAs were designated based in part on biological surveys, using those same data to evaluate the performance of ASPAs is circular. Even worse, ASPAs may be the subject of more surveys because they are ASPAs. Whatever the reason, bias in biological survey data could be bringing optimism into your analysis of ASPA biodiversity coverage. SDMs won't solve this completely, but they do go part way by extrapolating your data beyond the survey points. The case for this is well made in the literature. To be more specific, an analysis based on SDMs would at least allow you to see whether there are other parts of the island likely to contain the values that you claim to be limited to ASPAs.

Adding this aspect to the study does not mean that the SDMs analysis would replace or over-ride the current analysis based on point data – especially if the results of both approaches were qualitatively similar. It could be put into Supp material to assure readers that the impact of possible biases in the point data have been minimized, and that the qualitative conclusions arising from the point data align with outcomes of a state-of-art SDM analysis. Note that SDM analysis need not be used for all species, it is common to combine point data and SDMs in complementarity analyses for species that do not produce good SDMs.

I must apologise for recommending this extra work (which the Editor may choose to ignore), but I believe that if you want this work to have high policy impact, you should make sure it is as rigorous as it can reasonably be given the data at hand.

These comments are not made to try to underplay the importance of the work you have done already, or the relevance of your study to policy, but simply to add the level of rigour I see as necessary to your science and policy objective.

We thank Reviewer 2 for these additional comments. We would like to reiterate that these analyses represent one component of a suite of potential inputs that will inform the future development of area protection in Antarctica. We agree wholeheartedly that the use of Species Distribution Models (SDMs) represents an important mechanism for further assessing the representativeness of the current system and for identifying important areas for considerations in future ASPA designations. We have been actively exploring

these options within our research group for some time, and with the increasing availability of better resolution landscape and climate data, combined with the final version of the SCAR Antarctic Biodiversity Database, are confident that we will make significant advances over the next 12 months. Our preliminary work confirms that there are complex and diverse issues associated with these large scale SDM analyses in Antarctica and that adequate consideration of them is beyond the scope of this manuscript. In this context we believe that submitting them as an adjunct to this study would not allow the true value of SDMs to be appreciated and ultimately presented with maximum impact to the Antarctic Treaty system.

Therefore we maintain our position that the comprehensive occurrence data used here and the associated analyses and results, represents an important advance in understanding the current status of ASPA coverage, and for significantly progressing discussions on these issues in the Antarctic Treaty system.

In response to the specific points raised by this reviewer:

- i) We did not actually cite the work of Moirano et al. and believe the references cited following our statement regarding complexity (Ferrier and Guisan 2006 and Scherrer et al 2018) are appropriate. In particular the latter reference is useful in that it does cover, in some detail, the complexities of combining separate species distribution models, an aspect that we would see as vital to explore in any comprehensive Antarctic SDM analyses. We have added more text to this paragraph (Lines 93-97) to provide more detail on some of the specific complexities around this approach in terrestrial Antarctica.
- ii) We have now ensured that biases in the data are clearly identified in the text, adding extra text to note the potential for ASPAs to be better surveyed than non-protected areas (Lines 216-217).
- iii) We have also added text to reiterate the potential for SDMs to further inform the assessment and designation current and future area protection (Lines 217-221).